# The Influence of Agrotechnological Tools on cv. Rubin Apples Quality

Kristina Laužikė *, Nobertas Uselis and Giedrė Samuolienė

Institute of Horticulture, Lithuanian Research Centre for Agriculture and Forestry, Kauno 30, Kaunas Distr., LT-54333 Babtai, Lithuania; nobertas.uselis@lammc.lt (N.U.); giedre.samuoliene@lammc.lt (G.S.)
* Correspondence: Kristina.Lauzike@lammc.lt

**Abstract:** With the growing demand for quality food in the world, there is a new ambition to produce high-quality apples seeking reduced cultivation costs. The aim of this study was to evaluate the influence of agrotechnological tools on the quality of cv. Rubin apples during the harvest. The apple tree (*Malus domestica* Borkh.) cv. Rubin was grafted on dwarfing rootstocks P60, planted in single rows spaced 1.25 m between trees and 3.5 m between rows. Six agrotechnological tools were used—hand pruning, mechanical pruning, trunk incision, calcium-prohexadione, summer pruning and mechanical pruning one side, changing sides annually. The agrotechnical tools had no significant effect on accumulation of most sugars and elements, malic, folic and succinic acids in the fruits. Mechanical pruning resulted in significant accumulation of phenolic compounds, antioxidants, ascorbic acid, but reduced the amount of glucose compared to hand pruning. However, the trunk incision or spraying with ca-prohexadione together with mechanical pruning had no significant effect on sugar content but resulted in significantly higher amounts of phenols, antioxidants, ascorbic acid, Fe and Mn and reduced starch and citric acid.

**Keywords:** *malus domestica*; apples; carbohydrates; organic acids; quality





## 1. Introduction

Apples are one of the most widely grown and consumed fruits worldwide. Although the apple harvest area has started to decrease since 2015, the yield of apples grown remains the same—83–86 million tons per year [1].

Decreased cultivation area and the increased fruit demand raise the challenge for horticulture to produce an abundant and stable apple yield per land unit area. One way to reduce the cost of apple cultivation is to use mechanical pruning. Mechanical pruning reduces manual labor without losing [2,3]. Besides, the quality of apple fruits is even more important. Apples are rich in substances that reduce the risk of cardiovascular disease, asthma, diabetes and even some cancers and are assessed for their nutritional stability during storage and nutritionally valuable chemical composition [4–6]. Apples represent one of the most nutritional foods in a healthy diet, due to sugars (sorbitol, fructose, glucose, sucrose), organic acids (0.2–0.8%), vitamins (mainly vit. C, 2.3–31.1 mg/100 g DM) and water (>80%) contents [7–9]. The sweetness of the fruit is a big part of the consumers' choice, the good ratio of sugars and acids gives the consumer a pleasant taste [8,10,11]. Another important part of fruit quality are micro (zinc (Zn), iron (Fe), manganese (Mn), copper (Cu)) and macro (calcium (Ca), magnesium (Mg), potassium (K), sodium (Na)) elements [12–14].

Literature data prove that the chemical composition and quality of the apple is influenced by its variety [15–17], rootstock [18–20] and cultivation conditions (like weather, orchard system and other) [19,21]. Moreover, the quality of apples is strongly influenced by environmental factors. To obtain more uniform quality fruit, forming small, rare canopies is key. A smaller canopy reduces quality variation; the larger the canopy, the more uneven the quality of apples is [22,23]. Photosynthesis depends on light penetration through the

canopy and results in the synthesis of glucose leading to the accumulation of sorbitol in apple leaves, which are further transported through the phloem to fruit tissues. These sugars in fruits are converted depending on the plant developmental stage into fructose, glucose, malic acid, or starch [8,24]. In addition to light penetration into canopy, the supply of water and nutrients is very important for the fruit quality [25,26]. Water deficit decreases sorbitol accumulation in apple leaves, which is transported from the leaves to the fruit during fruit ripening, so drought stress also has a negative effect on the quality of the fruit [27,28]. Trunk incision disrupts the transport of water into the leaves by disrupting photosynthetic activity [3]. The amount of total nutrients stored in the fruit can be estimated from the amount of stored dry mass. Dry mass percentage in an apple depends on cultivar, rootstock, planting system, crop load, pruning technique and meteorological conditions [29–31].

Thus, the aim of this paper was to evaluate the effect of agrotechnological tools on the changes in fruit biochemical composition and other qualitative indices in cv. Rubin apples.

## 2. Materials and Methods

### 2.1. Plant Material and Growth Conditions

A trial was carried out in the experimental intensive orchard in Lithuania, ($55°60'$ N, $23°48'$ E) in 2017–2019. The apple tree (*Malus domestica* Borkh.) cultivar Rubin was grafted on dwarfing rootstocks P60. Apple trees were planted in 2010 in single rows spaced 1.25 m between trees and 3.5 m between rows. Pest and disease management was carried out according to integrated plant protection practices and orchard was not irrigated. The soil conditions of the experimental orchard were the following: clay loam, pH 7.3, humus 2.8%, $P_2O_5$ 255 mg kg$^{-1}$, $K_2O$ 230 mg kg$^{-1}$. Three single trees were fully randomized. The samples were collected from the whole canopy using full randomization at harvest time on commercial ripening. Randomly, 5 apples were taken from one replicate, finely chopped (after removing the seed boxes with the seeds) and part of fresh material was frozen in liquid nitrogen and the other parts dried for 48 h at 70 °C for elemental analysis. Six agrotechnological tools were used: 1. Each year hand pruning forming slender spindle (control); 2. Mechanical pruning (each year); 3. Trunk incision using chain saw + mechanical pruning (each year); 4. Mechanical pruning (each year) + spraying with calcium-prohexadione; 5. Mechanical pruning + hand pruning + summer pruning (performed in the middle of August removing the most vigorous and vertical shoots); 6. Mechanical pruning one side, changing sides annually. Agrotechnical tools have been applied every year since 2016 to the same orchard area. By mechanical pruning, the trees are pruned in the form of a trapezoid: horizontally—at a height of 0.7 m, the distance from the trunk is 0.6 m; at the height of 2.5 m–0.4 m, vertically, the trees were cut at a height of 2.5 m.

### 2.2. Determination of Soluble Sugars by UPLC

Soluble sugar (fructose, glucose, sorbitol) contents were evaluated using the UPLC method with evaporative scattering detection (ELSD) [32]. About 0.5 g of fresh plant tissue was ground and diluted with deionized water. The extraction was carried out for 4 h at room temperature, centrifuged at $14,000\times g$ for 15 min. A cleanup step was performed prior to the chromatographic analysis: 1 mL of the supernatant was mixed with 1 mL 0.01% (*w:v*) ammonium acetate in acetonitrile and incubated for 30 min at +4 °C. After incubation, samples were centrifuged at $14,000\times g$ for 15 min and filtered through 0.22 μm PTPE syringe filter (VWR International, Radnor, PA, USA). Analysis was performed on Shimadzu Nexera (Tokyo, Japan) system. Separation was performed on a Supelcosil $250 \times 4$ mm $NH_2$ column (Supelco, Bellefonte, PA, USA) using 77% acetonitrile as the mobile phase at 1 mL min$^{-1}$ flow rate. Calibration method was used for sugar quantification (mg g$^{-1}$ in FW).

### 2.3. Determination of Organic Acid by HPLC

Organic acid (oxalic, malic, citric and succinic) contents were determined using reversed phase HPLC method [33] on Shimadzu 10A (Japan) system with DAD. Sample was prepared grinding plant material and diluting with $H_2O$ 1:10 (*w:v*). Extraction was performed in heated water bath (50 °C) for 30 min. Extract was clarified by centrifugation at 10,000 rpm for 15 min and filtered through 0.22 μm PTPE syringe filter (VWR International, USA). Separation was performed on Lichrosorb RP-18 4.6 × 250 mm, 55 μm column (Altech). Mobile phase—0.05 M sulphuric acid, flow rate 0.5 mL min$^{-1}$, injection volume—10 μL. Calibration method was used for organic acid quantification (mg g$^{-1}$ in FW).

### 2.4. Determination of Micro- and Macro- Elements by ICP—OES Spectrometer

The macro and micro elements contents in apples were determined by microwave digestion technique combined with inductively coupled plasma optical emission spectrometry [34,35]. A complete digestion of dry apples (whole apple) material (0.5 g) was achieved with 100% $HNO_3$ using microwave digestion system Multiwave GO (Anton Paar GmbH, Graz, Austria). The digestion program was as follows: (1) 150 °C reached within 3 min, digested for 10 min; (2) 180 °C reached within 10 min, digested for 10 min. The mineralized samples were diluted to 50 mL with deionized water. The elemental profile was analyzed by ICP–OES spectrometer (Spectro Genesis, SPECTRO Analytical Instruments, Kleve, Germany). The operating conditions employed for ICP–OES determination were 1300 W RF power, 12 L min$^{-1}$ plasma flow, 1.0 L min$^{-1}$ auxiliary flow, 0.8 L min$^{-1}$ nebulizer flow, 1.0 mL min$^{-1}$ sample uptake rate. The analytical wavelengths (nm) chosen were: B I 249.773 nm, Ca II 445.478 nm, Cu I 324.754 nm, Fe II 259.941 nm, K I 766.491 nm, Mg II 279.079 nm, Mn II 259.373 nm, Na I 589.592 nm, P I 213.618 nm, S I 182.034 nm, Zn I 213.856 nm. The calibration standards were prepared by diluting a stock multi-elemental standard solution (1000 mg L$^{-1}$) in 6.5% (*v/v*) nitric acid, and by diluting a stock phosphorus and sulfur standard solutions (1000 mg L$^{-1}$) in deionized water. The calibration curves for all the studied elements were in the range of 0.01–400 mg L$^{-1}$.

### 2.5. Determination of Total Starch by Calorimetric Method

The total starch content was determined using the total starch Megazyme assay kit, a total starch assay kit based on the use of thermostable a-amylase and amyloglucosidase (Megazyme International Ireland Limited, Wicklow, Ireland), the method of determination of starch in samples, which also contains D-glucose and/or maltodextrins.

### 2.6. Determination of Total Phenolic Compounds by Calorimetric Method

Using a calorimetric method, the total content of phenolic compounds was determined using methanol extracts—1 g of plant tissues grounded with liquid nitrogen and diluted with 10 mL of 80% methanol. The extract was mix and left for 24 h in the fridge (+4 °C) and subsequently centrifuged at RCF4000 for 5 min; 0.1 mL of extract was diluted with 0.2 mL 10% Folin-Ciocalteau reagent (Folin reagent diluted with bi-distilled water 1:10) and with 0.8 mL 7.5% $Na_2CO_3$ solution [36]. The absorbance was measured after 20 min at 765 nm using a Genesys 6 spectrophotometer (Thermospectronic, Waltham, MA, USA) against distilled water as a blank. Gallic acid was used as a standard; the total phenolics were evaluated using a calibration curve.

### 2.7. Determination of DPPH Free Radical Scavenging Activity by the Calorimetric Method

The antioxidant activity of methanol extracts of the investigated plants was evaluated spectrophotometrically relating to the 2,2–diphenyl–1–picrylhydrazyl (DPPH) free radical scavenging capacity. One gram of plant tissues grounded with liquid nitrogen and diluted with 10 mL of 80% methanol. The extract was mix and left for 24 h in the fridge (+4 °C) and subsequently centrifuged at RCF4000 for 5 min. One mL DPPH solution (60 μ DPPH) and 0.1 mL of extract add into cuvette. The absorbance scanned after 16 min from the beginning of the reaction at 515 nm [37,38].

*2.8. Determination of the ABTS Radical Scavenging Activity by Calorimetric Method*

The ABTS (2.2'-azino-bis (3-ethylbenzothiazoline-6-sulfonic acid) diammonium salt) scavenging activities of apples extracts were determined. ABTS was dissolved in methanol at a concentration of 2 mM. The ABTS radical cation was produced by incubating the ABTS stock solution with 200 μL $K_2S_2O_8$ (0.1982 g/10 mL $H_2O$) in the dark for 16 h. Following this, 100 μL of the diluted sample was mixed with 2 mL of ABTS solution and the absorbance was scanned for 11 min (plateau phase) at 734 nm. The ABTS scavenging activity of each extract was calculated as the difference between the initial absorbance and that after reacting for 10 min, which was expressed as mmol (ABTS) scavenged per 1 g fresh sample (mmol $g^{-1}$). Methanol was used as the blank solution [39].

*2.9. Statistical Analysis*

MS Excel Version 2010 and XLStat 2020 Data Analysis and Statistical Solution for Microsoft Excel (Addinsoft, Paris, France) statistical software were used for data processing. Analysis of variance (ANOVA) was carried out along with Turkey multiple comparisons test for statistical analyses, $p \leq 0.05$ ($n + 9$, 3 years, 3 repetitions each). Multivariate principal component analysis (PCA) was performed. The results are presented in PCA scatter plot that indicate distinct levels according to all analyses performed.

*2.10. Meteorological Conditions*

The meteorological data were collected from "iMetos" meteorological station at Institute of Horticulture, LAMMC, Lithuania. During the vegetation period, the mean temperature was close to perennial for all three years of the experiment (Table S1). Meanwhile, precipitation was irregular, rainy during fruit ripening.

**3. Results**

Agrotechnological tools did not have a significant effect on the amount of fructose and sucrose in the fruit, but significantly affected the the amount of glucose, sorbitol and starch (Figure 1). Hand pruning resulted in the highest amount of glucose in the apples. Mechanical pruning with or without other tools resulted in the decrease of glucose 50–42% compared to hand pruning. Compared to hand (1) and mechanical (2) pruning, accumulation of starch decreased up to 27% in trunk incision (3), 46% with ca-prohexadione (4), up to 49% in summer pruning (5) and up to 67% mechanical pruning one side (6) treatments, respectively.

In contrast to carbohydrate accumulation, hand pruning (1) resulted in a significant decrease of total phenols and antioxidant activity (Figure 2). Mechanical pruning (2) and mechanical pruning with hand pruning (5) increased total phenols up to 1.2 mg $g^{-1}$ FW compared to hand pruning (1). All treatments with mechanical pruning increased DPPH and ABTS radical scavenging activity. DPPH radical scavenging activity increased up to 9–13%, meanwhile DPPH radical scavenging activity increased up to 24–46%. The ABTS free radical scavenging activity method was more sensitive than DPPH and showed more significant differences. Trunk incision (3) and ca-prohexadione (4) significantly decreased ABTS radical scavenging activity, but not as much as other mechanical pruning treatments.

The major organic acid found in cv. Rubin fruits was malic acid (3.8–4.0 mg $g^{-1}$ FW), followed by oxalic, oxalacetic and succinic acids (0.11–0.14 mg $g^{-1}$ FW), ascorbic (17–18 μg $g^{-1}$ FW) and citric acids (8–19 μg $g^{-1}$ FW), folic (1–2 μg $g^{-1}$ FW) and fumaric acids (2–3 μg $g^{-1}$ FW). The most significant differences were found in the amounts of ascorbic and citric acids (Figure 3). Compared to hand pruning, all mechanical pruning treatment combinations (2–6) increased the accumulation of ascorbic acid in the fruit up to 11%–15%. Citric acid contents were 33%–36% bigger in hand (1) and mechanical (2) pruning compared with mechanical pruning combinations. The same tendency, but no significant, was observed for oxalic acid accumulation. No significant difference between the treatments was observed for other organic acids (malic, oxalacetic, succinic, folic and fumaric).

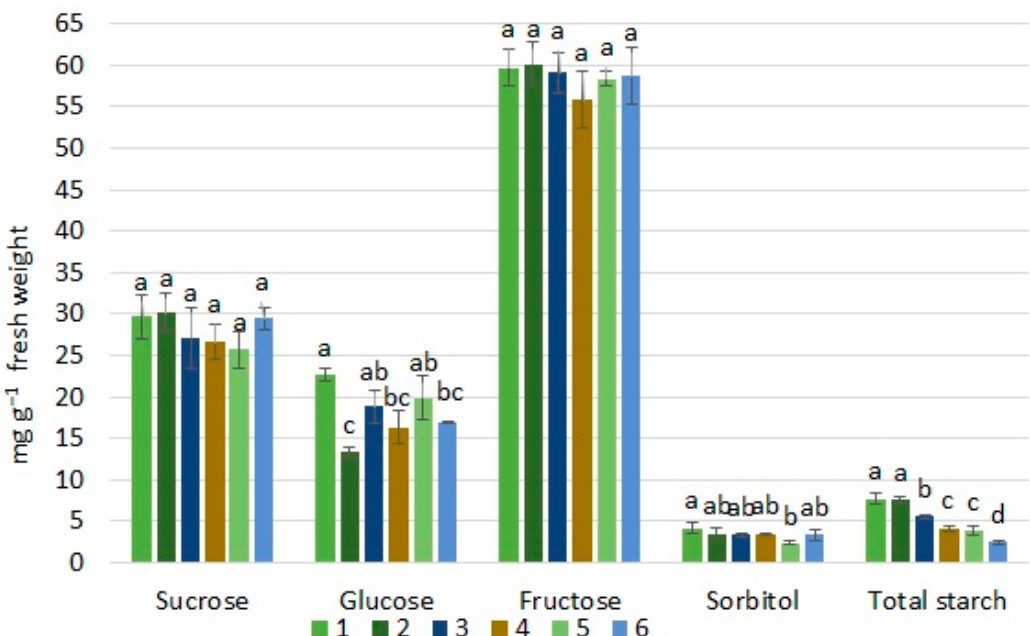

**Figure 1.** Influence of agrotechnological tools on carbohydrate content in cv. Rubin apples on harvest time (commercial ripening) in average of three years. 1. Each year hand pruning forming slender spindle (control); 2. Mechanical pruning (each year); 3. Trunk incision using chain saw + mechanical pruning (each year); 4. Mechanical pruning (each year) + spraying with calcium-prohexadione; 5. Mechanical pruning + hand pruning + summer pruning; 6. Mechanical pruning one side, changing sides annually. Averages followed by different letter within the same compound indicate significant differences according to the Duncan's least significant difference test (*p* < 0.05). Error bars show standard deviation.

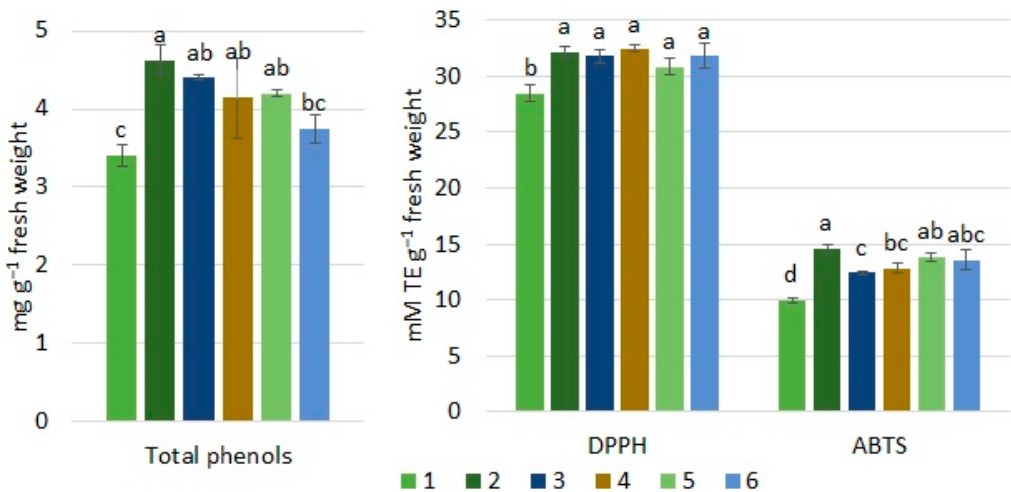

**Figure 2.** Influence of agrotechnological tools on total phenols, the 2,2–diphenyl–1–picrylhydrazyl (DPPH) and ABTS (2.2′-azino-bis (3-ethylbenzothiazoline-6-sulfonic acid) diammonium salt) radical scavenging activity in cv. Rubin apples on harvest time (commercial ripening) in average of three years. 1. Each year hand pruning forming slender spindle (control); 2. Mechanical pruning (each year); 3. Trunk incision using chain saw + mechanical pruning (each year); 4. Mechanical pruning (each year) + spraying with calcium-prohexadione; 5. Mechanical pruning + hand pruning + summer pruning; 6. Mechanical pruning one side, changing sides annually. Averages followed by different letter within the same compound indicate significant differences according to the Duncan's least significant difference test (*p* < 0.05). Error bars show standard deviation.

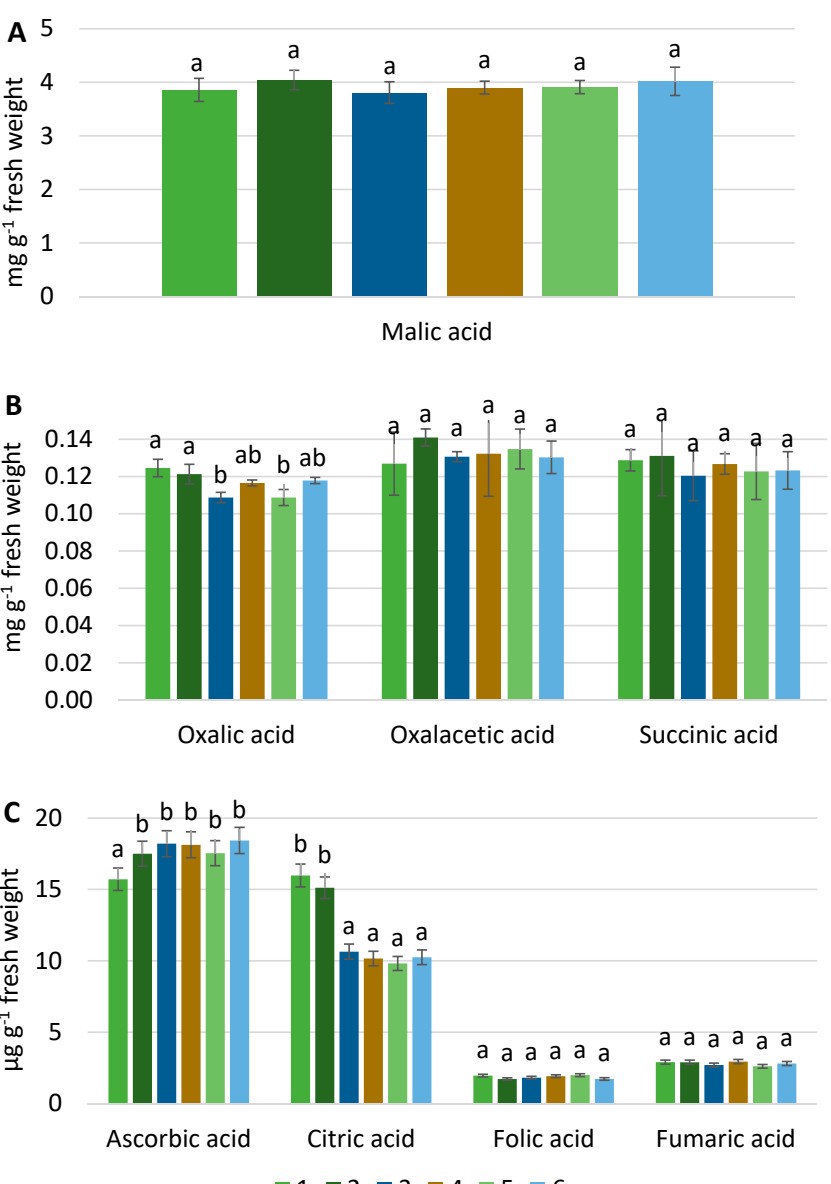

**Figure 3.** Influence of agrotechnological tools on organic acids (**A**), (**B**), (**C**) in cv. Rubin apples on harvest time (commercial ripening) in average of three years. 1. Each year hand pruning forming slender spindle (control); 2. Mechanical pruning (each year); 3. Trunk incision using chain saw + mechanical pruning (each year); 4. Mechanical pruning (each year) + spraying with calcium-prohexadione; 5. Mechanical pruning + hand pruning + summer pruning; 6. Mechanical pruning one side, changing sides annually. Averages followed by different letter within the same compound indicate significant differences according to the Duncan's least significant difference test ($p < 0.05$). Error bars show standard deviation.

The content of macro and micro elements in cv. Rubin fruits was in the following order: K (0.76–0.92 mg g$^{-1}$ FW), Ca, Mg, Na (6–25 mg g$^{-1}$ FW), Fe (2–3 µg g$^{-1}$ FW), Mn (0.8–1.0 µg g$^{-1}$ FW), Cu (0.4–0.7 µg g$^{-1}$ FW), and Zn (0.1–0.6 µg g$^{-1}$ FW) (Figure 4). The applied technological tools did not affect the accumulation of Mg and Na. Hand pruning (1) resulted in a significant decrease of all K, Ca and all microelements (Figure 4A). Mechanical pruning (2) led to a significant decrease of Fe and Zn as well (Figure 4B). Other combinations with mechanical pruning had positive effect especially on K, Fe Cu and Zn accumulation.

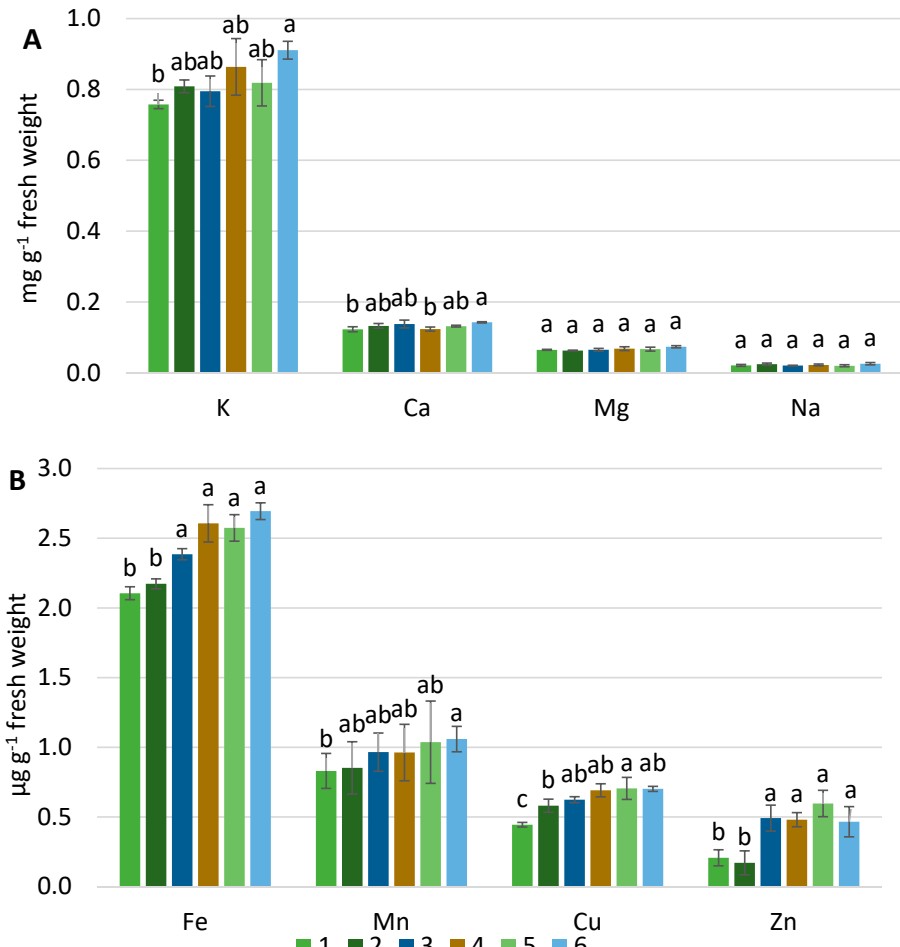

**Figure 4.** Influence of agrotechnological tools on major (**A**) and trace (**B**) elements in cv. Rubin apples on harvest time (commercial ripening) in average of three years. 1. Each year hand pruning forming slender spindle (control); 2. Mechanical pruning (each year); 3. Trunk incision using chain saw + mechanical pruning (each year); 4. Mechanical pruning (each year) + spraying with calcium-prohexadione; 5. Mechanical pruning + hand pruning + summer pruning; 6. Mechanical pruning one side, changing sides annually. Averages followed by different letter within the same compound indicate significant differences according to the Duncan's least significant difference test ($p < 0.05$). Error bars show standard deviation.

The PCA scatterplot show an average coordinate of carbohydrates, phenolic compounds, antioxidant activity, macro and micro elements in cv Rubin fruits when apple trees were treated (with hand pruning, mechanical pruning and mechanical pruning with additional tools (Figure 5). The first two factors (F1 vs. F2) of the PCA, explained 49.30% of the total variance in response to agrotechnological tools. F1 explained approximately 33%, whereas F2 explained 16.6% of the total variability. In terms of F1 score, the plant responses to applied mechanical pruning with different combinations clearly distinct from the responses to hand pruning.

Using manual pruning apples accumulates higher amounts of carbohydrates but less phenolic compounds and weaker antioxidant activity, as well as lower amounts of elements (Figure 6). Trunk incision or spraying with ca-prohexadione together with mechanical pruning maintains sugar content as hand pruning, significantly increases levels of phenols, ascorbic acid, Fe, Mn and antioxidant activity, while at the same time reduces starch and citric acid.

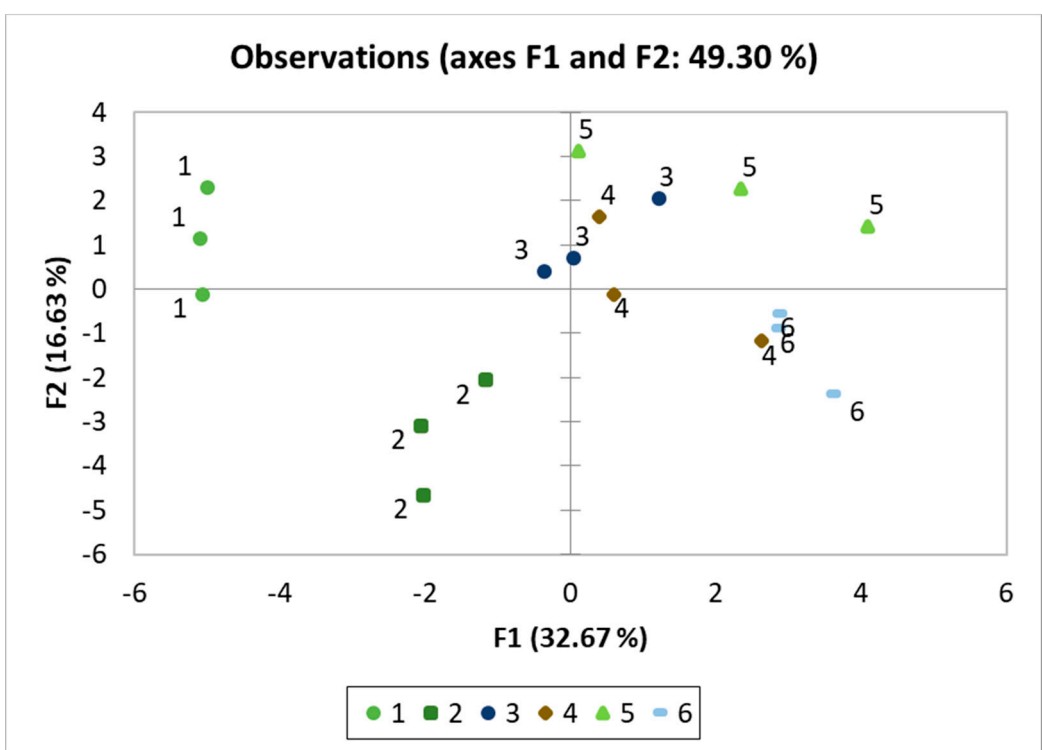

**Figure 5.** The principal component analysis (PCA) scatterplot, indicating distinct differences in carbohydrates, phenols DPPH and ABTS radical scavenging activity, organic acids and elements in cv. Rubin apples on harvest time (commercial ripening) in average of three years. 1. Each year hand pruning forming slender spindle (control); 2. Mechanical pruning (each year); 3. Trunk incision using chain saw + mechanical pruning (each year); 4. Mechanical pruning (each year) + spraying with calcium-prohexadione; 5. Mechanical pruning + hand pruning + summer pruning; 6. Mechanical pruning one side, changing sides annually.

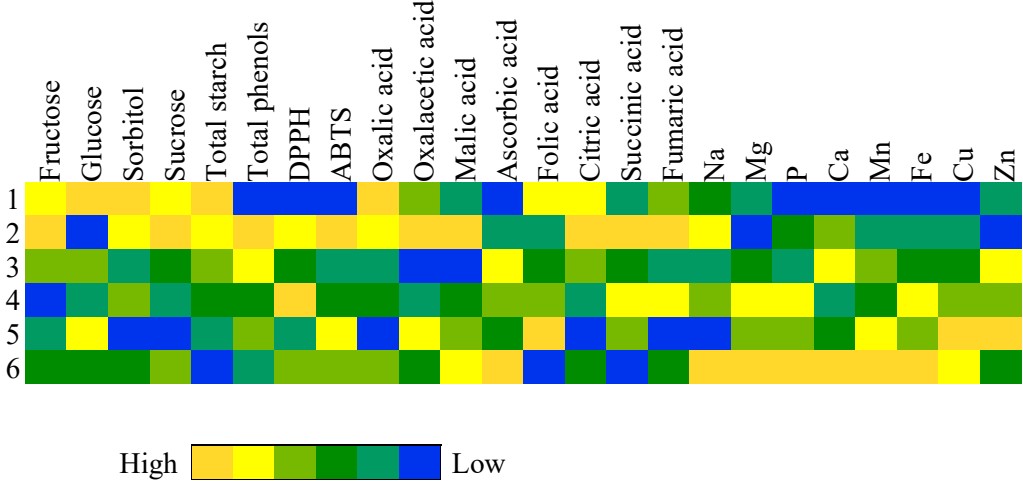

**Figure 6.** Relationships of agrotechnological measures according to compound concentrations in cv. Rubin apples on harvest time in average of three years. 1. Each year hand pruning forming slender spindle (control); 2. Mechanical pruning (each year); 3. Trunk incision using chain saw + mechanical pruning (each year); 4. Mechanical pruning (each year) + spraying with calcium-prohexadione; 5. Mechanical pruning + hand pruning + summer pruning; 6. Mechanical pruning one side, changing sides annually.

## 4. Discussion

Hand pruning positively affected the accumulation of most carbohydrates but negative effect on antioxidant activity and elements accumulation was observed (Figure S1). Apples grown by hand pruning accumulated the most sugar but less antioxidants and elements.

Apples grown with manual pruning accumulated mainly sugar, but less antioxidants and elements.

Meanwhile, using only mechanical pruning, the sugar content did not change, only the glucose was reduced, but the highest antioxidant activity, the content of phenolic compounds and the number of elements was increased compared to hand pruning (Figure S1).

Our previous studies have shown that mechanical pruning reduced the photosynthetic productivity of cv. Rubin apples leaves, fruit size, but significantly increased yield [3]. In this study, we analyze the influence of mechanical pruning on fruit quality. Biddlecombe and Dalton [40] stated that the significant effect of mechanical and hand pruning on the fruit sugar content remains only in the first year of cultivation, after which these accumulated sugar levels equalize and no significant differences remain. Our data partially confirms this statement. Regardless of the pruning method used, no significant differences were found between amount of sucrose, fructose and sorbitol in the cv. Rubin fruit (Figure 1). However, only mechanical pruning, even in fully grown trees, results in a significantly lower accumulation of glucose in cv. Rubin fruits compared to hand pruning. One of the reasons for the lower glucose accumulation in the fruit may be the reduced photosynthetic activity due to the dense canopy [3]. Mechanical pruning had no significant effect on fruit starch content in our study, and the same data were provided by Frazen and Hirst [41]. However, additional tools, such as trunk incision or spraying with ca-prohexsadione in combination with mechanical pruning significantly reduced cv Rubin fruit starch content, as did summer pruning. Trunk incision and ca-prohexadione reduced shoot size for cv. Rubin [42], the fruit were better exposed to the sun, and also in combination with summer pruning, the fruit got more sun during ripening. In agreement with previous statement cv. Rubin fruits with summer pruning resulted in better ripening and less starch in the fruit (Figure 1). Summer pruning in our study did not have a significant effect on Rubin fruits, only significant reduction of starch was observed. Tahir et al. [43] showed, that summer pruning improved fruit color and shelf life without compromising yield.

Pruning causes stress to the apples and increase the number of phenols [44]. The increase in phenolic compounds indicates that mechanical pruning caused bigger stress to the apple trees compared to hand pruning (Figure 2). During mechanical pruning, all branches of the tree that fall within the cutting edge are cut down, while during manual pruning, only certain branches are cut without damaging others. Mechanical pruning led to increased antioxidant activity as well (Figure 2). Viškelis et al. [45] found that apple tree cv. Rubin canopy formation methods increased phenol content compared to hand pruning. In agreement with Viškelis et al. [45], ca-prohexadione resulted in slightly lower accumulation of phenols compared to mechanical pruning, but it was higher compared to hand pruning. However, these differences were not significant for accumulation of total phenols. According to Drogoudi and Pantelidis [46], fruits getting more sun and their skins accumulate more phenols, but light does not affect the phenols and antioxidants in the fruit flesh. Thus, it can be presumed that the increase in phenols accumulation was not caused by light but was a result of stress caused by agrotechnical tools especially mechanical pruning without other tools (Figure 2).

Feng et al. [47] studied three apple tree varieties and found that accumulation of organic acids depended on the fruit position in the canopy—exposed to the sun or to the shade. However, contradictory data shows that fruit shading does not affect the number of organic acids [47,48]. Our data shows that in contrast to ascorbic acid accumulation, mechanical pruning in complex with trunk incision, praying ca-prohexadione significantly reduced the accumulation of oxalic and citric acids in cv. Rubin fruits (Figure 3).

In agreement with Biddlecombe and Dalton [40] mechanical pruning did not affect the number of elements accumulated in the fruits (Figure 4). However, the elemental composi-

tion of the fruits changes significantly using other agrotechnological tools together with mechanical pruning. Trunk incision or spraying with ca-prohexadione resulted in a significant accumulation of Fe and Zn (Figure 4) and these data are confirmed by Kviklys et al. (2020), where they found that treatments with trunk incision and ca-prohexadione increased the amount of iron up to 22.4% and 25.5%, respectively. These changes improve the composition of the fruit by the number of elements that are beneficial to the processes of the human body. These changes improve the composition of the fruit by the number of elements that are beneficial to the processes of the human body.

These changes will improve the fruit composition of elements, which are beneficial to the human body processes [49,50].

## 5. Conclusions

Summarizing all the obtained results, it can be stated that mechanical pruning not only reduces hand work, but also has a positive effect on cv. Rubin fruits. Mechanical pruning results in a significant increase of phenolic compounds (up to 35%), ascorbic acid (up to 11%) and antioxidant activity (up to 12–46%) but reduces the accumulation of glucose more than 70% compared to hand pruning. However, trunk incision or spraying with ca-prohexadione together with mechanical pruning maintains sugar content, significantly increase levels of phenols, ascorbic acid, Fe, Mn and antioxidant activity, at the same time reduce starch and citric acid. Thus, mechanical pruning with trunk incision or spray ca-prohexadione can result in significantly better cv. Rubin fruit quality.

**Supplementary Materials:** The following are available online at https://www.mdpi.com/2073-4395/11/3/463/s1, Table S1: Average monthly temperature and precipitation over three years and multiannual (100-year average).

**Author Contributions:** Investigation, writing—original draft preparation, visualization, K.L.; methodology, installation and maintenance of the experiment, N.U.; writing—review and editing, supervision, G.S. All authors have read and agreed to the published version of the manuscript.

**Funding:** This research received no external funding.

**Institutional Review Board Statement:** Not applicable.

**Informed Consent Statement:** Not applicable.

**Acknowledgments:** This work was carried out within the framework of long-term research program "Horticulture: agrobiological basics and technologies" implemented by Lithuanian Research Centre for Agriculture and Forestry.

**Conflicts of Interest:** The authors declare no conflict of interest.

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
