# Peer review of "The Influence of Agrotechnological Tools on cv. Rubin Apples Quality"

_agronomy, doi:10.3390/agronomy11030463_

Round 1

Reviewer 1 Report

The paper The influence of agrotechnological tools on cv. Rubin apples quality present an interesting study on improving the quality of apples using modern techniques. More over the information are related with the research made on this topic in the last years. However, for publication the authors must made some improvements:

Page 1, line 21 (Keywords): Replace “malus domectica” with “malus domestica

Page 1, line 21 (Keywords): Replace “carbocydrates” with “carbohydrates”

Page 1, line 31 (Introduction): Replace “Applesare” with “Apples are”

Page 1, lines 41-42 (Introduction). There are other factors that influence the chemical composition and quality of apples. These factors can be both natural (e.g. pedology, geological substratum) and anthropogenic (e.g. pesticide usage, nearby industrial pollution sources, road traffic). If you referred to these factors when talking about environmental factors, please specify in parentheses.

Page 2, line 45 (Introduction): Replace “frui” with “fruit”

Page 2, Materials and Methods (Plant material and growth conditions). I consider that it is necessary to introduce additional information related to the surface of the orchard, the number of samples taken and analyzed (total, from each tree). It is not clearly explained how the six agrotechnological tools were applied in the orchard.  Each tool has been applied to a specific area of the orchard during all the three years?

Page 3, Materials and Methods (Determination of micro- and macro- elements by ICP – OES spectrometer). The microwave digestion technique was carried out on whole or peeled fruits?

Page 3, line 126 (Materials and Methods): Replace “filtered filtered” with “filtered”

Page 3, line 126 (Materials and Methods): Replace “frigde” with “fridge”

Page 3, line 133 (Materials and Methods): Replace “Determinatio” with “Determination

Page 3, line 137 (Materials and Methods): Replace “frigde” with “fridge”

Page 4, Materials and Methods (Statistical analysis). How many samples were analyzed? The PCA method was used but is not presented in the subchapter of statistical analysis.

Page 8, Line 248. Replace “(Fig. 6)” with “(Fig. 5)”

Page 8, Figure 5. I suggest that the authors use the same colors for the six agrotechnological tools as in the previous figures.

Page 8, line 255 (Fig. 5 caption): Replace “organic acids end elements” with “organic acids and elements”

Page 9, line 328 (Conclusions): Replace “ca-prohexacione” with “ca-prohexadione”

Figure captions show the influence of agrotechnological methods on apple composition on harvest time. Please specify that the fruits were harvested at full maturity (at commercial ripening or full physiological one?).

If there is no limit number of figures imposed by the journal, I consider that Figure S1 can be included in the article as Figure 6.

Please check that the References chapter is formatted according to the instructions of the journal.

Regarding this observations, we consider that the paper can be published after the authors will made the modification required.

Author Response

Dear reviewer,

First, we would like to thank you for your very precise and clear comments and your attention to spelling.

All spelling mistakes have been corrected, thanks again for the comments.

Page 1, line 41-42. We supplemented and specified these effects on fruit quality.

Page 2, Materials and Methods (Plant material and growth conditions). We supplemented information on sampling and preparation, as well as the use of agrotechnical tools.

Page 3, Materials and Methods (Determination of micro- and macro- elements by ICP – OES spectrometer). The microwave digestion technique was carried out on whole fruits, we supplemented the method information.

Page 4, Materials and Methods (Statistical analysis). All samples were taken for three years, each time in triplicate, and 9 samples were used for statistical analysis. The section on statistical analysis has been supplemented with the necessary information.

Page 8, Figure 5. Thanks for the suggestion, we adjusted the Figure 5 and compared the colors to the other figures. As well as figures supplemented with information on apple ripeness. Apples were picked at a technical maturity - commercial ripening.

We agreed to suggestion to include Figure S1 int the article as Figure 6, thank You very much for 

Reviewer 2 Report

I haven’t any knowledge about the methods used to evaluate the different quality parameters in apples. I have no experience in pruning trials on apple trees.  I can only evaluate the aspects related to the pruning methodology as well as see the general aspect of the results presentation and discussion.

The description of agrotechnological tools used must be increased. The treatments used, specially the treatments with mechanical pruning should indicated the type of cut made (horizontal or vertical) and the distance to the tree. A mention of the machine used increased the paper quality. Also an indication of the pruning season for the mechanical interventions. A description of summer pruning should be included.

In the discussion, authors mentioned the greater stress caused by mechanical pruning. A better explanation of this stress could be useful.

Figure 5 shows a PCA analysis result, but any reference to this analysis was included in the point 2.9 – Statistical analysis. For readers who are unaware of PCA analysis, should explain better what allows to obtain.

The text have some jackdaws and should be revised by a native english speaker. Some examples:  Line 31, line 50, line 263, line 266, line 271, line 297, line 304. The reference to table S1 in line 159. That table doesn’t exist.

An increase in the quality of graphics should be consider. It include the Y axis and a legend to distinguish the treatments used from the title of the figure. A better pattern of the columns could be used in order to increase the quality of the figures.

Author Response

Dear reviewer,

Thank you very much for taking the time to review the manuscript, as well as for your critical and prestigious comments.

We increased the descriptions of agrotechnical tools and their using. Also, specified the mechanical pruning.

About the stress. During mechanical pruning, all branches of the tree that fall within the cutting edge are cut down, while during manual pruning, only certain branches are cut without damaging others. Larger tree damage and reduced light penetration due to a compacted crown may be the cause of increased tree stress.

The section on statistical analysis has been supplemented with the information about PCA and samples.

All spelling mistakes have been corrected and checked.

Table S1 is in supplementary material. We do not consider it necessary to encumber the manuscript with this information, but it is necessary to provide it as additional material.

Sorry I didn't fully understand your suggestions for figures.

Round 2

Reviewer 1 Report

The Authors thoroughly replied to all comments raised by the revision. The paper, now, is surely ameliorated. In this form, it can be considered for publication.